# Phylogenetic and Evolutionary Studies of Grapevine Pinot Gris Virus Isolates from Canada

**DOI:** 10.3390/v15030735

**Published:** 2023-03-12

**Authors:** Minh Vu, Bhadra Murthy Vemulapati, Wendy McFadden-Smith, Mamadou L. Fall, José Ramón Úrbez-Torres, Debra L. Moreau, Sudarsana Poojari

**Affiliations:** 1Cool Climate Oenology and Viticulture Institute, Brock University, St. Catharines, ON L2S 3A1, Canadabvemulapati@brocku.ca (B.M.V.); 2Ontario Ministry of Agriculture, Food and Rural Affairs, Vineland Station, Lincoln, ON L0R 2E0, Canada; wendy.mcfadden-smith@ontario.ca; 3Saint-Jean-sur-Richelieu Research and Development Centre, Agriculture and Agri-Food Canada, Saint-Jean-sur-Richelieu, QC J3B 7B5, Canada; 4Summerland Research and Development Centre, Agriculture and Agri-Food Canada, Summerland, BC V0H 1Z0, Canada; joseramon.urbeztorres@agr.gc.ca; 5Kentville Research and Development Centre, Agriculture and Agri-Food Canada, Kentville, NS B4N 1J5, Canada; debra.moreau@agr.gc.ca

**Keywords:** grapevine, high-throughput sequencing, phylogenetic analysis, virus evolution, data mining

## Abstract

This study investigated the phylogenetic relationship of grapevine Pinot gris virus (GPGV) isolates from Canada with GPGV isolates reported worldwide. Full-length genomes of 25 GPGV isolates representing the main four grape-growing regions in Canada (British Columbia, Ontario, Nova Scotia and Quebec) were sequenced and compared to genomes of 43 GPGV isolates representing eight countries and three continents. Phylogenetic analysis based on full genome sequences revealed an unambiguous separation of North American GPGV isolates with isolates from Europe and Asia. Within the North American clade, GPGV isolates from the USA segregated into a distinct subclade, whereas the relationships amongst GPGV isolates from different regions of Canada were not clearly defined. The phylogenetic analysis of the overlapping regions of MP and CP genes involving 169 isolates from 14 countries resulted in two distinctive clades, which were seemingly independent of their country of origin. Clade 1 included the majority of asymptomatic isolates (81% asymptomatic), whereas clade 2 was predominantly formed of symptomatic isolates (78% symptomatic). This research is the first study focused on the genetic variability and origin of GPGV in Canada.

## 1. Introduction

In 2003, a novel disease that caused leaf mottling, leaf deformation and poor vigor was first observed in vineyards around the Trentino region in Italy. The disease was named grapevine leaf mottling and deformation (GLMD). Infected vines presented fewer canes and clusters, with a significant decrease in cluster weight [1]. In 2012, a novel virus was discovered in this region and named grapevine Pinot gris virus (GPGV) after the variety in which it was found [2]. It was not until 2019 that GPGV’s causal role in GLMD was established for the first time [3]. GPGV was reported in different *Vitis vinifera* cultivars from various grape-growing regions such as Algeria [4], Armenia [5], Argentina [6], Australia [7], Belgium [8], Brazil [9], Bulgaria [10], Canada [11], Chile [12], China [13], France [14], Germany [15], Greece [16], Iran [17], Italy [16], Japan [18], Lebanon and the Middle East [19], Moldavia [20], Pakistan [21], Poland [22], Slovakia and the Czech Republic [23], Slovenia [24], South Korea [25], Spain [26], Turkey [27] and the USA [28].

GPGV is a positive-sense single-stranded RNA virus belonging to the genus *Trichovirus*, family *Betaflexiviridae* [2]. The viral genome is approximately 7250 nucleotides with three overlapping open reading frames (ORFs), which encode for replicase-associated proteins, movement protein (MP) and coat protein (CP) [2]. The genome of GPGV shares significant similarities with grapevine berry inner necrosis virus, (GINV) and both viruses evoke comparable symptoms on leaves and shoots, although GPGV was never found to cause any symptoms in berries [29]. Unlike GINV, whose presence was confirmed only in Japan [30] and China [31], the wide spread of GPGV poses a serious threat to the grape and wine industry globally, prompting a number of studies on the origin and symptomology of its strains [32,33].

GPGV is graft-transmissible among *Vitis* species [29]. However, the severity of infection and symptom development in different cultivars remains uncertain. In the Trentino region of Italy, for example, cultivar Teroldego appears to be resistant to GLMD [29]. Tocai and Glera, on the other hand, developed severe symptoms after being grafted with GPGV-infected rootstocks [34,35]. However, it was shown that within the same cultivars (Pinot gris, Traminer and Pinot noir), infected vines can show a wide range of symptomology, ranging from severely symptomatic to asymptomatic, with a comparable prevalence (95% and 87.1% of symptomatic and asymptomatic vines infected with GPGV, respectively) [29,36].

Due to its uncertain symptomology, the assessment of the economic impact of GPGV is quite challenging. Until now, there have only been a few attempts to associate symptomology with genetic variations of different GPGV isolates through phylogenetic analysis. One of the earliest studies reported the presence of a non-synonymous single nucleotide polymorphism (ns-SNP) at the 3′-end of the MP gene of the virus, which causes a nucleotide change from cytosine into thymine by substituting a glutamine residue with a stop codon. Glutamine was usually present in the MP gene of GPGV isolates from asymptomatic grapevines (latent), whereas the premature stop codon was commonly detected in the MP gene of GPGV isolates from asymptomatic grapevines (virulent), which showed a shortening of the MP gene by six amino acids [37]. This ns-SNP was found to be important but not decisive in symptom expression [37]. Since then, the region surrounding this ns-SNP (which will henceforth be referred to as MP/CP) has become a prime target for phylogenetic analysis. Analysis of the MP/CP region from 41 GPGV isolates from Pinot gris and Glera cultivars in Northern Italy revealed that GPGV could be segregated into three groups, one of which is comprises predominantly symptomatic isolates [36]. Furthermore, in 2019, a full genome analysis of 20 GPGV isolates from a variety of grape cultivars (Pinot gris, Glera, Tocai Friulano, Tannat, Merlot, Riesling, Vetliner and Touriga Nacional) and an herbaceous host (*Silene latifolia*) with various countries of origin (Italy, Uruguay, France, Canada, the USA, Germany and Slovakia) revealed that they could be segregated into at least two groups, one of which is exclusively composed of asymptomatic isolates [33]. It was not until 2021 that the putative involvement of genetic variability on symptom expression was experimentally demonstrated by Tarquini et al. [38]. In their study, a chimeric clone was synthesized by replacing a 356 bp segment at the end of the MP gene of a symptom-inducing strain (or virulent strain, fvg-Is12, accession number MH087443, Appendix A) with the same segment derived from a non-symptom-inducing strain (or latent strain, fvg-Is15, accession number MH087446, Appendix A). The chimeric clone behaved very similarly to the latent parent in terms of symptoms expression, virus titre, effect on gene expression and virus-derived siRNA. This result proved that the 3′-end of MP gene plays a crucial role in not just symptom expression, but in other pathological aspects of GPGV as well [38]. However, because this study is the first and only study of this nature, it is unclear if these 356 bp segments from fvg-Is12 and fvg-Is15 are representative of “virulent” and “latent” GPVG isolates, respectively.

The grape and wine industry in Canada contributes significantly to the national economy, with an annual economic impact of more than CAD$11 billion. Ontario (ON) is Canada’s largest and most important viticulture area, contributing CAD$5.5 billion dollars per year (numbers from Wine Growers Canada and Wine Growers Ontario, 2019). According to a survey conducted in 2018, around 22% of grapevines from targeted vineyards in Ontario tested positive for GPGV [39]. The presence of GPGV was also confirmed recently in other regions of Canada such as British Columbia (BC) [11], Quebec (QC) [40] and Nova Scotia (NS) [41].

Considering the prevalence of GPGV in Canada and the potential threat that it poses, the main of objectives of this study were to report the relative incidence of GPGV in ON and BC and to investigate the genetic variability of Canadian GPGV isolates, their origin and their relationships to GPGV isolates from other countries. The secondary objective was to assess if the difference in the end of the MP region fvg-Is 12 (allegedly virulent parent) and fvg-Is15 (allegedly latent parent) is representative for other GPGV isolates as well.

## 2. Materials and Methods

### 2.1. Plant Material

#### 2.1.1. Relative Incidence

To estimate the incidence rates of GPGV in ON and BC, samples were collected from randomly chosen vineyards in their respective provinces. The number of samples from each variety was structured to best represent the demographic of grapevine variety in each region. A total of 1352 composite samples were collected (884 from Ontario and 468 from BC). Each composite sample represents 20 mature petioles collected from 5 vines. Four mature leaves covering both sides of the lower canopy were chosen to ensure comprehensive coverage of the unevenly distributed virus titers. Samples were collected in September and October of 2020 and 2021 regardless of the presence/absence of GPGV and the severity of symptoms displayed by grapevine plants. Total nucleic acids (TNAs) were extracted from these samples to test for the presence of GPGV, and endpoint-PCR was performed with the primer set targeting the coat protein gene [34]. The prevalence of GPGV was calculated by number of positive samples in RT-PCR divided by the total number of samples and was reported in Table 1.

#### 2.1.2. Sample Selection for High Throughput Sequencing

Based on the survey data, 21 GPGV-positive *V. vinifera* samples from seven different cultivars were selected as candidates for HTS. From each individual vine, four mature leaves from both sides of the trellis, thereby covering the entire canopy, were collected. Of the twenty-one samples, thirteen were from ON, seven were from BC and one was from NS. Vines that showed clear leaf mottling and deformation symptoms were assigned the status “symptomatic”, whereas vines that did not show any of those symptoms were assigned the status “asymptomatic” (Table 2).

#### 2.1.3. Total RNA Extraction

Leaf tissues were ground with liquid nitrogen, and the total RNAs were extracted using the Spectrum™ Plant Total RNA Kit (Sigma-Aldrich, Oakville, ON, Canada) by following the manufacturer’s instructions. A quality check was performed with the NanoDrop 1C platform (Thermo Fisher—Mississauga, ON, Canada). Samples with an A260/A280 ratio of 1.8 or higher were used in subsequent analysis.

### 2.2. High Throughput Sequencing (HTS)

Twenty-one samples with desired levels of quality subsequently underwent HTS library preparation using the TruSeq^®^ Stranded Total RNA Library Prep Plant (96 samples) and TruSeq DNA CD Indexes (96 indexes, 96 samples) (Illumina, San Dieago, CA, USA) by following the manufacturer’s instructions. All samples were pooled and sequenced using the Miseq-Illumina platform at CCOVI, Brock University, Canada. Approximately one million pair end reads on average were generated for each sample (average size ranging from 116.9 to 137.7 bp—Table 3).

### 2.3. Data Analysis

#### 2.3.1. Assembling Reads and Recovering GPGV Genome

Two separate procedures were deployed in tandem to analyze the generated reads. The first procedure aimed to detect all plant viruses and viroids present in each sample using Virtool [42] (www.virtool.ca (accessed on 1 October 2022)). The reads imported to Virtool were trimmed of adapters and low-quality reads (minimum Q20) with Skewer 0.2.2 [42], and their quality was assessed with FastQC 0.11.5+ (https://www.bioinformatics.babraham.ac.uk/projects/fastqc/ (accessed on 1 October 2022)). Reads of sufficient quality were then subjected to a scan via alignment using PathoScope [43] (a computational framework based on Bowtie 2 2.2.3+ [44]) with a customized plant viruses and viroids database derived from GenBank [45]. Any viruses or viroids with more than 15 percent coverage or bearing more than 1000 matching reads were considered positive [45]. Virtool also reported the depth of the sequence recovered, which measures how many times the recovered genome was covered by mapped reads. Generally, Virtool’s read mapping approach is superior to the contig assembly approach in terms of sensitivity and speed due to the fewer reads that it requires [35]. To extract consensus sequences for virus species, CLC Genomics Workbench 20.0.4 (CGW) was used. The reads imported to CGW were trimmed with the threshold of Q20 (automatic Illumina’s adapters detection; maximum ambiguity is 2; retain homopolymers if present at both ends; minimum read length of 20 and a maximum read length of 151). Grapevine genome sequences were eliminated using the 19 chromosomes of the grape sample PN40024 as reference (NCBI Accession numbers NC_012007.3 to NC_012025.3). The sequence of GPGV in each sample was then extracted via reference (NC_015782.1) assembly. The 21 GPGV full genomes recovered were annotated using NCBI-ORF-Finder and submitted to NCBI (accession number from OK558797 to OK558817, Table 2). Additionally, five assembled genomes of GPGV were provided by Dr. Fall’s lab at AAFC Saint-Jean-sur-Richelieu RDC (Table 2).

#### 2.3.2. Obtaining Additional Data

To expand our data set, in addition to the 26 GPGV isolates obtained in this study, we included 38 GPGV isolates with complete or near-complete genomes available in NCBI whose country of origin and symptoms expressed were available, resulting in a total data set of 64 isolates. Data regarding the identities, accession numbers, symptoms, cultivars and references of the additional sequences are given in Appendix A.

#### 2.3.3. Recombination Analysis

The 64 full genomes were first aligned with the MUSCLE algorithm in MEGA-X using default parameters. The extent of recombination in the sequences objected in this study was assessed with both SplitTree 5.3 (preliminary visual assessment via a neighbor joining network, Figure 1A) and RDP4 (breakpoint predictions via full exploratory recombination scan using RDP, GENECONV, Bootscan/Rescan, Chimaera, MaxChi, SisScan and 3seq methods, Appendix A). Breakpoints were considered significant if it was confirmed by five or more statistical methods. A total of 7 significant breakpoints were found (Appendix A).

#### 2.3.4. Phylogenetic Analysis of Full Genome and Recombination Free Segment

To select the most appropriate phylogenetic model, we relied on the MEGA-X “Find best model” function (initial tree: Neighbor Joining, statistical method: Maximum likelihood—ML). Based on the model suggested, a rooted, bootstrap consensus maximum likelihood tree (bootstrap value of 1000) was constructed for the 64 full genome sequences alongside an outgroup. Because the accuracy of the rooted tree is improved with the closeness of the outgroup compared to the ingroups [39], apple chlorotic leaf spot virus (ACLSV), which belongs to the same *Trichovirus* genus as GPGV, was chosen as the outgroup (NCBI Acc. # NC_001409). From the seven breakpoints, eight recombination-free segments were derived (Appendix A). The largest segment was 4866 bp (position 2293 to 7159 in alignment, Appendix A), allowing meaningful phylogenetic analysis to be done. A neighbor joining network (Splitree 5.3, Figure 1B) and a rooted ML tree (MEGA-X, Figure 3B) was constructed for this segment using the protocol described previously. Intraclade and interclade distances for the two rooted ML trees were estimated with MEGA-X using the Maximum Composite Likelihood model (Appendix A). No preference for start codon position was selected. All ambiguous positions were removed for each sequence pair (pairwise deletion option). To better visualize the extent of recombination in our data set, a consensus network of ML trees (unrooted, 1000 bootstrap) from segments with lengths of 200 bp or more was constructed. The unrooted ML tree for the largest segment can be found in Appendix A, and their consensus network is shown in Figure 2. The rooted ML tree for full genome and the 4866 bp fragment is shown in Figure 3A,B respectively.

#### 2.3.5. Phylogenetic Analysis of the MP-CP Region

To investigate if the differences at the 3′-end of the MP region between the parents of this chimeric strain, namely fvg-Is12 (virulent parent) and fvg-Is15 (latent parent), is representative of virulent and latent GPGV isolates, respectively, we analyzed a segment of 460 bp spanning the 3′-end of MP gene and the beginning of CP gene from 168 GPGV isolates found in plants whose symptoms were clearly recorded (21 sequences from the present study and 147 sequences from NCBI). From the 168 isolates, 73 were symptomatic, and 95 were asymptomatic (see Appendix A). A maximum likelihood model for these 168 sequences was constructed using the protocol described previously (Figure 4). The distances between clades and within each clade of the MP/CP model were estimated with MEGA-X using the Maximum Composite Likelihood model. No preference for start codon position was selected. All ambiguous positions were removed for each sequence pair (pairwise deletion option) (Appendix A).

## 3. Results

### 3.1. Relative Incidence

The prevalence of GPGV (positive samples/total samples) was significantly higher in ON (34.3%) than in BC (9.8%). In fact, all of the 12 cultivars that were sampled in ON were infected with GPGV. The cultivars with the highest GPGV incidence rates in ON were Sauvignon blanc (65%), Merlot (61.7%) and Pinot noir (50%). In contrast, only six out of seventeen cultivars sampled from BC were found positive for GPGV. Other than the three cultivars in BC with the highest incidence rates (Auxerrois 65%, Siegerrebe 48.1% and Gamay noir 40%), the other cultivars all had relatively low incidence rates (less than 15%) (Table 1).

### 3.2. GPGV Isolates and HTS Sequencing

Twenty-one libraries were sequenced simultaneously, generating 1 million reads on average per library. Details regarding their names, cultivars, origins and accession numbers can be found in Table 2. After trimming off adapters and low-quality reads (Q20 threshold), approximately 1.8% of the total reads were removed. Most of these reads belonged to the grapevine genome (ranging from 74% to 99%, with an average of 97%). After grapevine genome subtraction, 1% to 5% of reads from each library belonged to GPGV (with an average of 2.4% of reads). A more detailed breakdown of the raw sequencing data can be found in Table 3. Twenty-one complete genomes of GPGV were recovered from those twenty-one libraries by reference-assembling the reads on NC_015782. The depth (ranging from 4 to 95 times) and coverage (ranging from 95–100%) of each assembled genome is summarized in Table 3 and is shown in Appendix A.

### 3.3. Virus Detection Using Virtool

Grapevine Pinot gris virus was detected in all the samples. Most noticeably, Virtool also detected eight incidents of grapevine leafroll-associated virus 3 (GLRV-3) and a single incident of grapevine red blotch virus (GRBV). Furthermore, hop stunt viroid (HSVd), grapevine rupestris stem pitting-associated virus (GRSPaV) and grapevine associated tymo-like virus (GaTLV) were found in most samples. Grapevine Syrah virus 1 (GSyV-1) and grapevine fleck virus (GFkV) were detected twice with relatively low coverages. Grapevine leafroll-associated virus 1 (GLRaV-1) and grapevine virus T (GVT) were each detected only once. A summary of virus and viroid species detected other than GPGV can be found in Table 4.

### 3.4. Recombination Analysis

The initial neighbor joining network shows significant reticulation within BC isolates as well as within Italian isolates, indicating that recombination events between isolates of the same clade may occur (Figure 1A). The exploratory scan with RDP4 revealed a total of eight recombination events, four of which were supported by five or more of the methods tested, resulting in seven significant breakpoints (Appendix A). These events were found in BC5, BC8, NS-1, fvg-Is7, fvg-Is 8, fvg-Is12, fvg-Is14, fvg-Is17 and FEM01 isolates (Appendix A). Not all GPGV isolates were extracted from *Vitis* species, except for isolate FEM01, which was extracted from *Silene latifolia*, an herbaceous host [46]. The *p*-values, which represent the probability of the null hypothesis for these events in each detection method, are summarized in Appendix A. From the seven breakpoints detected, eight recombination-free segments were derived, with three larger segments (more than 200 bp) and four smaller segments (less than 200 bp) (Appendix A). Three ML trees for each of the larger segments were constructed. The ML tree for the largest segments can be found in Appendix A (Appendix A).

The neighbor joining network of the largest recombination-free fragment (4866 bp) showed a similar overall topology to the full genome network (Figure 1A,B). However, a significant reduction of reticulation around BC and Italian isolates was observed, allowing better assessment of their local genealogy.

In the consensus network of the three ML trees from the larger segments, a highly intricate grid manifested around Italian isolates, indicating contradictory phylogenetic signals from each segment. Although a grid-like structure was also observed around BC isolates, it was of significantly lower complexity (Figure 2).

### 3.5. Phylogenetic Analysis on Full Genome and Recombination-Free Segment

The maximum likelihood model rooted on ACLSV divided 64 GPGV isolates into two larger clades (clades 1 and 2) and two smaller clades (clades 3 and 4) (Figure 3A). The largest clade, clade 1, included 43 isolates, which were almost exclusively isolates from North America (42/43), except for isolate Mer from France. Clade 2 contained mostly European isolates (10/12), except for isolates S142 and 12G1110, which were from North America. It was noticeable that clade 2 was more closely related to clade 1 than to other clades found in this model. Clades 3 and 4 contained exclusively isolates from Europe. Isolate SL13 from Pakistan did not belong to any of the clades found here, which may be due to its Asian origin.

A closer look at clade 1 revealed the single isolate from Europe (Mer from France) to be genetically very close to isolate BC-1 from Canada, an observation supported by the 100% bootstrap value. Within clade 1, all isolates from the USA were grouped together and formed a tight subclade, whereas the relationships between Canadian isolates from different provinces were not as clear. For clades 2, 3 and 4, except for the isolates from Slovakia (in clade 2), there was no clear separation between isolates of different geographical origins. Regarding the expression of symptoms, clade 4 stood out with the highest symptomatic/asymptomatic ratio of 83% (5/6 isolates), followed by clade 3 (1/2 or 50%). Clades 1 and 2 had lower symptomatic/asymptomatic ratios of 19% (8/43) and 25% (3/12), respectively.

The ML model for the 4866 bp fragment retained a similar overall topology, dividing 64 GPGV isolates into four clades (Figure 3B). Clades 1 and 2 were identical to those from the full genome model. The Fvg-Is8 isolate shifted from clade 3 to clade 4, whereas SL13 was still not included in any clade. The exclusion of fvg-Is8 from clade 4 resulted in a clade with 100% isolates (5/5) from symptomatic vines. The overall improvement in bootstrap consensus was attributable to the elimination of recombination as well as the contradicting signals that it entails.

### 3.6. Phylogenetic Analysis of the MP/CP Region

The ML tree of 168 MP/CP overlapping regions contained two major clades. The first clade, named α, (Figure 4, green-shaded), represented 99 isolates, and the second clade, named β (Figure 4, red-shaded), represented 69 isolates. Under closer inspection, clade α contained isolates from North America, South America, Europe and Africa. Clade β contained almost exclusively isolates from Europe with the exception of isolates SL-13 (from Pakistan), ME and CF (from Brazil). Within clade 1, all isolates from the USA grouped together and formed a complete subclade. This pattern was not observed with isolates from any other country. Regarding symptoms, a significant portion of clade 1 was made up of isolates from asymptomatic vines (80/99 isolates or 80%), whereas clade 2 was mostly composed of isolates from symptomatic vines (54/69 isolates, or 78.2%) (Figure 4). The fvg-Is12 isolate, collected from symptomatic Pinot gris grapevine, was found in clade β, whereas the fvs-Is15 isolate, which was isolated from asymptomatic Pinot gris grapevine, was included in clade α (Figure 4).

## 4. Discussion

Discovered in 2012, GPGV has since been reported in all major grape growing regions around the world, including Canada [11,40,41]. The economic importance of GPGV is hard to estimate due to the wide range of severity of the symptoms it can induce. Previous works showed a connection between genetic variability and symptom induction by GPGV. However, there has been no study on the genetic variability of GPGV in Canada. Furthermore, an analysis on the relationship of GPGV isolates found in Canada with GPGV isolates worldwide would also have significant implications for disease management and the sourcing of propagative material.

The relative Incidence rate of GPGV in ON (34.3%) was shown in this study to be significantly higher than the previous estimation of 21.6% by Xiao et al. in 2018 [31]. British Columbia, on the other hand, showed a much lower incidence rate (9.8%) than ON. Poojari et al. (2020) [41] reported low incidence (4.6%) of GPGV from NS vineyards. From the survey results in this study, there is no clear pattern of infection based on cultivars. In ON, white and red *V. vinifera* cultivars showed relatively the similar incidence rates, whereas hybrids appeared to have lower incidence rates (Table 1). In contrast, hybrids in BC and NS had significantly higher incident rates than white and red *V. vinifera* cultivars [41].

In this study, 21 GPGV-infected vines from ON, BC and NS were selected to undergo total RNA sequencing with Illumina-Miseq, and full GPGV genomes were obtained with an average coverage of 99% and a depth of 17.8 (Table 2). Other than GPGV, ubiquitous grapevine viruses and viroids were also detected, namely GRSPaV (17/21 samples), HSVd (17/21 samples) and GYSVd (5/21 samples) (Table 3). Moreover, GLRaV-3 and GaTLV were both detected with relatively high frequency (in 8 and 10 samples, respectively), which agreed with previous studies that reported the high prevalence of GLRaV-3 in BC [47], ON [39] and NS [41]. GSyV and GFkV were each detected twice while GRBV, GLRV-1 and GVT were all detected once (Table 3). The fact that 100% of the vines were found to be infected with at least two viruses confirmed the extent of the coinfection of grapevines in Canada.

The relationship between GPGV isolates identified in Canada and those reported worldwide was assessed via phylogenetic analysis. A total of 64 GPGV isolates from eight countries and three continents were included in the study. Because viruses frequently engage in evolutionary processes that might violate the assumptions of a simple sequence substitution model, recombination analysis was necessary. An initial neighbor joining network of the 64 GPGV genomes resulted in a dense grid amongst Italian isolates, indicating the presence of recombination amongst these isolates (Figure 1A). This pattern of reticulation was observed previously by Tarquini et al. in 2019 [33]. Furthermore, a few interbranch linkages amongst BC isolates as well as ON isolates were also found (Figure 1A). An exploratory scan with RDP4 provided similar patterns predicting three significant recombination events amongst the Italian isolates and one between BC isolates (Appendix A), resulting in seven significant breakpoints (Appendix A). Eight recombination-free segments were derived between these breakpoints (Appendix A). Three of those fragments retained sufficient size to undergo meaningful phylogenetic analysis (Appendix A). The consensus network built from the unrooted ML model of those three fragments showed a highly intricate grid-like structure between the Italian isolates as well as some minor reticulation between BC isolates (Figure 2). Because the length of the edges in the consensus network is proportional to the frequency it was found in the constituent trees, a net-like structure indicates contradictory phylogenetic signals from each segment. These contradictory signals are attributable to recombination, which is not rare within the *Betaflexiviridae* family and were documented repeatedly [48,49]. For recombination to happen, coinfection of the same host with different virus strains is a prerequisite and usually happens with the help of a vector [50]. The only confirmed vector of GPGV to date, *Colomerus vitis* (commonly known as grape Erineum mite), was shown in a semi-controlled environment to be capable of transmitting GPGV [51]. If fed on vines infected with different strains of GPGV, *C. vitis* could cause coinfection, and it might subsequently lead to recombination. Because many herbaceous and woody species [24,46] are reported to host GPGV, there might be other factors contributing to spread and recombination in natural conditions.

To avoid misinterpretation of the evolutionary relationship between these isolates, two rooted ML trees were constructed, one using the full genome and the other using the largest recombination-free fragment. In addition to the anticipated improvement in bootstrap consensus due to recombination elimination, the two ML trees displayed almost identical topologies. Clades 1 and 2 from the two trees were identical, with clade 1 containing almost exclusively North American isolates (with the exception of isolate Mer from France) and clade 2 containing mostly European isolates (except for isolates S142 and 12G1110 from North America). Most notably, the recombination-free ML model manifested a clade in which 100% of isolates were found in symptomatic vines (Figure 1). This result agrees with the phylogenetic analysis of Tarquini et al. in 2019 [33]. The γ clade in their study, which was dubbed the “virulent” clade, is equivalent to our clade 4 with an additional isolate from a symptomatic host (PN from France), whereas their β clade is equivalent to our clade 3 with the addition of isolates Gr and SL13. The clades found in these analyses also mirror what was found in a study by Hily et al. in 2020 [32], in which 126 GPGV genomes were divided into three broad clades with origins mostly from France, Italy and China, respectively. Compared to their study, clade 1 of this study would fit in their French clade, clade 2 would fit in their Italian clade, and clades 3 and 4 would stand on the edge between the Italian and Chinese clades [32]. 

When evolutionary divergence was estimated, both the full genome tree and the recombination free tree demonstrated that the intraclade distance of clade 1 was significantly lower than those of clades 2, 3 and 4 (Appendix A). The tightness of clade 1 implies a recent common ancestor for GPGV isolates found in North America. An extensive evolutionary history analysis by Hily et al. suggested that GPGV originated from Asia, spread to Europe and then to North and South America via plant material importation [32]. Considering that the only European isolate that was associated with North American isolates in this study was isolate Mer from France, it is possible that the common ancestor of GPGV in North America has a French origin. This hypothesis is also consistent with the fact that the majority of plant material in Canada was imported from France. The 21 GPGV isolates from three distinct viticulture regions of Canada did not display complete geographical separation (Figure 1). For ON isolates, 8/13 isolates (ON-1, 2, 3, 4, 5, 6, 8 and 9) formed a robust subclade (99% and 92% bootstrap values in full genome and recombination-free trees, respectively). These isolates were more closely related to GPGV isolates found in the USA than to GPGV isolates found in other part of Canada. For BC isolates, three isolates were found to be closely related (BC 2, 3 and 4, bootstrap values of 100% and 99%). For all other Canadian isolates, including the five from QC, there were no apparent patterns of separation. This lack of separation implies that GPGV isolates from different provinces of Canada originated from closely related sources. Considering the distance between ON, BC, NS and QC, the propagation of infected material, rather than an insect vector, might be the main source of the dissemination of GPGV in Canada.

Regarding the correlation between genotype and symptoms induction by GPGV, there have been significant efforts to find a link between them. Notable insights include the discovery of an SNP in the MP gene that lead to an early truncation [37] as well as the establishment of the supposed virulent and latent clades [33,36]. However, a definitive link between genotype and symptoms induction was only established recently with the work of Tarquini and colleagues [38]. Their construction of chimeric strains unequivocally showed the importance of the 3′-end of the MP gene in GPGV’s etiology [38]. A phylogenetic analysis focused on this region was conducted with 168 GPGV isolates from 14 countries and 5 continents. The rooted ML tree divided the 168 GPGV isolates into two clades with contrasting proportions of symptom-inducing isolates. Clade α (the latent clade) represented 99 isolates with 80% (80/99 isolates) originating from asymptomatic vines; clade β (the virulent clade) represented 69 isolates with 78% (54/69) of them originating from symptomatic vines. Interestingly, the virulent parent (fvg-Is 12) and the latent parent (fvg-Is 15) in Tarquini et al.’s study fell into the virulent clade and latent clade, respectively. That means 74% (54/73) of the symptom-inducing isolates found in this study resembled fvg-Is 12 more closely than fvg-Is 15. On the other hand, 84% (80/95) of non-symptom-inducing isolates resembled fvg-15 more closely than fvg-Is12. Thus, the difference in MP/CP regions between these two isolates seems to be detectable at the phylogenetic level and is representative of the majority of GPGV isolates around the world.

Despite the clear correlation between genetic variability in the MP/CP region with symptom induction, a lack of association for a significant portion of the samples was evident. Indeed, 34/168 of the samples (20%) in the MP/CP analysis were found in the “unexpected” clade. Therefore, it seems that genetic differences between GPGV strains is not the only factor that affects the presence of symptoms. One important factor to consider is the titre of the virus. It was shown repeatedly for GPGV that virus titre can affect both symptom expression and virus detection [36,37]. Because we did not quantify the GPGV titre in the samples, this aspect remained unexamined. The timing of sample collection also plays a crucial part in symptom expression. During the growing season, virus titre can vary significantly, leading to variations in symptoms observed [36]. The duration of infection should also be considered. From the work of Tarquini et al. in 2019, healthy vines inoculated with the latent strain GPGV (fvg-Is15) via agrodrench with agrobacterium in greenhouse conditions showed obvious symptoms at the beginning of inoculation, but manifested symptom-free leaves for 4 months post-inoculation [3]. This phenomenon was also observed in healthy vines inoculated with the virulent strain (fvg-Is15), albeit the manifestation of asymptomatic leaves was significantly delayed (5 months post-inoculation) [3]. An antagonistic effect of co-infection with other viruses remains a possibility, but thus far, there is no evidence to support this hypothesis. In addition to ubiquitous viruses and viroids such as GRSPaV, HSVd and GYSV-1, grapevines are regularly infected with GFkV, GLRaV-1, GLRaV-3 and GRBV [41,47,52]. Based on the few previous studies which tested for the presence of other viruses, symptom induction by GPGV did not co-segregate with any virus [28,35,36]. The only symptom-inducing isolate in this study, ON-10, co-infected the host with GPGV, GRBV, GVT, HSVd and GRSPaV. Given that HSVd and GRSPaV were previously found to have no association with the symptom expression of GPGV [35,36], GRBV and GVT are the only candidates to be antagonistic coinfectors (Appendix A). The absence of GRBV and GVT in the other 20 non-symptom-inducing isolates is also consistent with this assumption. Nevertheless, neither GRBV nor GVT seem necessary for symptom expression when we consider a study by Al Rwahnih et al. in 2021 [28], who reported two asymptomatic vines harbouring both GPGV and GRBV, whereas all symptomatic vines were free of GVT. According to their study, GFLV was found as a co-infector with GPGV in all five symptomatic vines, whereas none of the asymptomatic vines were found infected with GFLV [25]. This observation, in turn, contradicted other studies in which vines without GLFV were still found to be symptomatic [33,36]. In addition to the viruses and viroids mentioned, grapevines can be infected with at least 80 more viruses or viroids, of which the majority are not economically important and are often overlooked and understudied. Thus, a complete rejection of the effects of co-infection on symptom expression would be rather speculative at this point.

## 5. Conclusions

In this first study of genetic variability of GPGV in Canada, it was found that GPGV isolates in Canada are closely related to GPGV isolates in the USA and are phylogenetically distinct from GPGV isolates from Europe and Asia. Considering the tightness of the North American clade as well as the suggested evolutionary history of GPGV from other studies [32,33], it is likely that GPGV isolates in North America share a recent common ancestor of French origin. Lastly, the MP/CP region of fvg-Is 12 is representative of the majority of symptom-inducing isolates, whereas the region of fvg-Is 15 is representative of the majority of non-symptom-inducing isolates. The difference in the MP/CP region amongst symptom-inducing and non-symptom-inducing isolates is detectable at a phylogenetic level. However, this difference is not the sole determinant of symptom expression, as other factors such as virus titre, sampling time and duration of infection also play important roles.

## Figures and Tables

**Figure 1 viruses-15-00735-f001:**
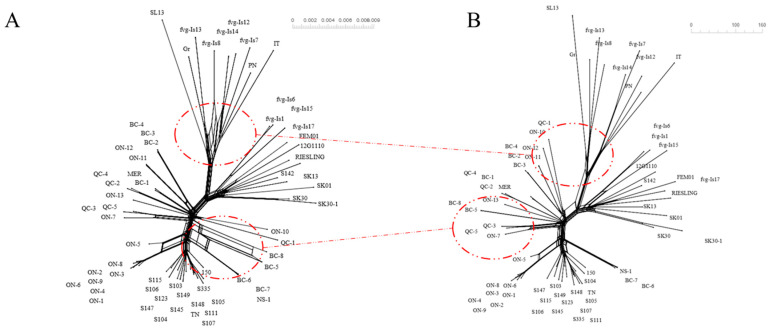
Initial neighbor joining network for 64 full genomes (approximately 7300 bp, (**A**)/left) and for the 64 largest recombination-free fragments (4866 bp, (**B**)/right). Both trees were built using the neighbor joining network pipeline in Splittree 5.3.0, which consists of the Hamming Distances method (Hamming 1950), the Neighbor Net method (Bryant and Moulton 2004) and the Splits Network Algorithm method (Dress and Huson 2004), using all default options.

**Figure 2 viruses-15-00735-f002:**
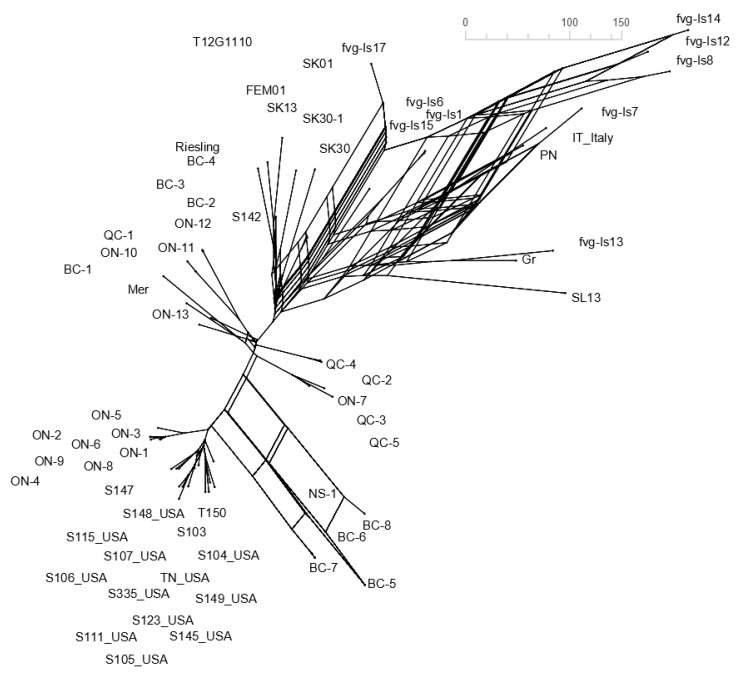
Consensus network built from three ML trees of larger recombination-free segments. The Consensus Network method (Holland and Moulton 2003) and the Splits Network algorithm method (Dress and Huson 2004) was used with default options and accepting threshold of 0.33 (Splittree 5.3.0). ML tree for the largest segment (4866 bp) is shown in Appendix A.

**Figure 3 viruses-15-00735-f003:**
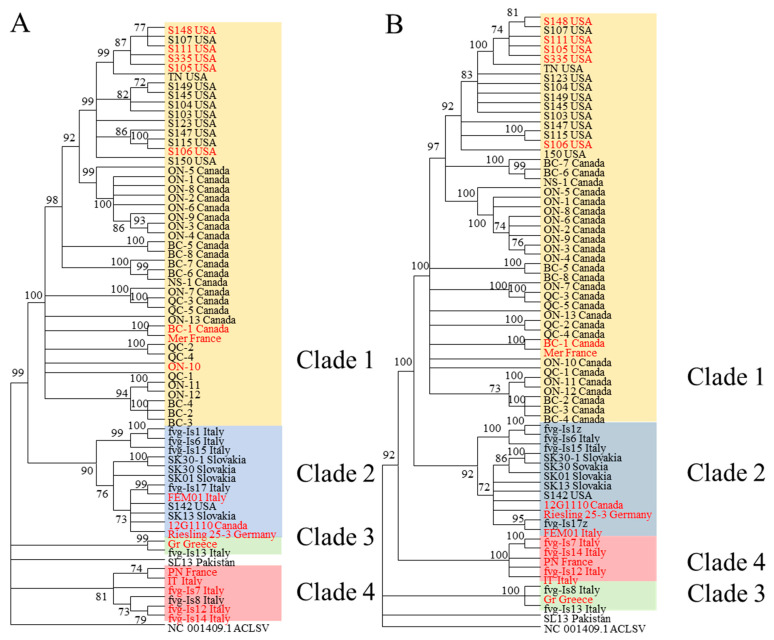
Maximum likelihood model for full genomes ((**A**)/left) and the largest recombination-free fragments 4866 bp ((**B**)/right) from 64 GPGV isolates. Both of these rooted bootstrap-consensus maximum likelihood trees were constructed with MEGA-X (General Time Reversal Model with discreet gamma distribution and invariant sites) and a bootstrap value of 1000, using ACLSV as an outgroup. Branches with less than 70% bootstrap consensus were collapsed. GPGV isolates collected from symptomatic grapevines are written in red.

**Figure 4 viruses-15-00735-f004:**
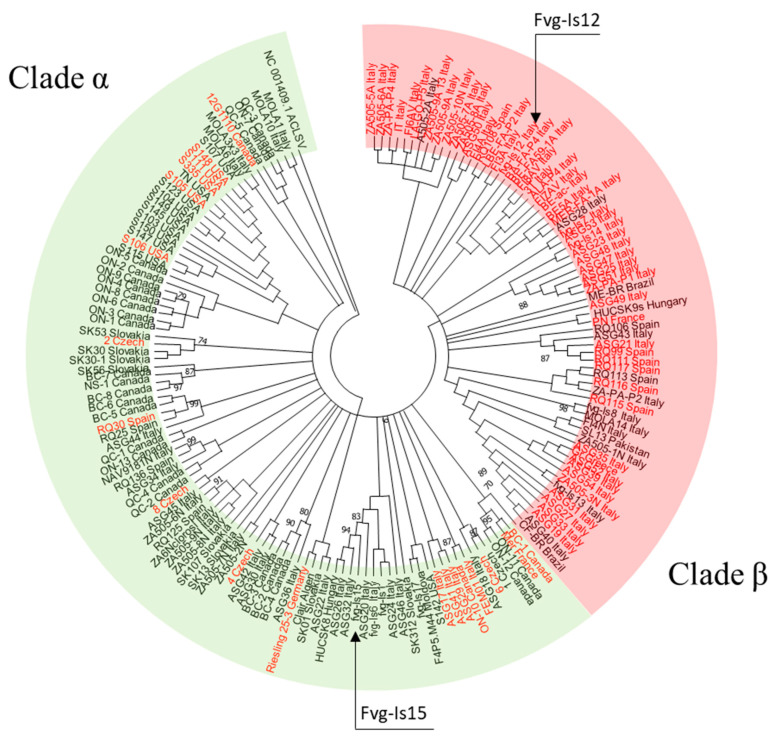
Phylogenetic relationships between 168 GPGV isolates regarding the MP/CP interregional sequence. This rooted neighbor joining tree was constructed with MEGA-X using a bootstrap value of 1000 and using apple chlorotic leaf spot virus as the outgroup. Nodes with higher than 70% bootstrap consensus are annotated. GPGV isolates collected from symptomatic grapevines are written in red. The green-shaded area indicates clade α, and the red-shaded area indicates clade β. A circular tree was selected instead of the traditional rectangular tree due to the high number of isolates.

**Table 1 viruses-15-00735-t001:** Survey results from ON and BC for GPGV in different cultivars.

Cultivar	Total # of Samples	# of Sample with GPGV	% Relative Incidence
**Ontario**			
*White viniferas*			
Chardonnay	100	38	38.0
Pinot gris	20	7	35.0
Chardonnay Musquet	20	4	20.0
Riesling	140	36	25.7
Sauvignon blanc	60	39	65.0
*Red viniferas*			
Cabernet franc	120	49	40.8
Cabernet sauvignon	80	27	33.8
Merlot	60	37	61.7
Pinot noir	40	20	50.0
*Interspecific hybrids*			
Baco noir	80	13	16.3
Vidal	164	33	20.1
**Total**	884	303	34.3
**British Columbia**			
*White viniferas*			
Chardonnay	29	0	0.0
Pinot gris	75	11	14.7
Riesling	40	0	0.0
*Red viniferas*			
Cabernet franc	30	0	0.0
Cabernet sauvignon	9	1	11.1
Gamay noir	20	8	40.0
Merlot	133	0	0.0
Pinot noir	69	0	0.0
*Interspecific hybrids*			
Auxerrois	20	13	65.0
Dornfelder	1	0	0.0
Kerner	1	0	0.0
La belle	2	0	0.0
Marechal Foch	4	0	0.0
Ortega	2	0	0.0
Petit Verdot	4	0	0.0
Siegerrebe	27	13	48.1
Spvigel	2	0	0.0
**Total**	468	46	9.8

**Table 2 viruses-15-00735-t002:** Information on the cultivars, symptoms, origins and accession numbers of 21 GPGV isolates sequenced in this study and the five assembled genomes of GPGV provided by Dr. Fall’s lab at AAFC Saint-Jean-sur-Richelieu RDC.

Isolate	Symptom	Cultivar	Origin	Accession Number
ON1	Asymptomatic	Chardonnay	Ontario	OK558797
ON2	Asymptomatic	Chardonnay	Ontario	OK558798
ON3	Asymptomatic	Chardonnay	Ontario	OK558799
ON4	Asymptomatic	Chardonnay	Ontario	OK558800
ON5	Asymptomatic	Chardonnay	Ontario	OK558801
ON6	Asymptomatic	Chardonnay	Ontario	OK558802
ON7	Asymptomatic	Chardonnay	Ontario	OK558803
ON8	Asymptomatic	Chardonnay	Ontario	OK558804
ON9	Asymptomatic	Chardonnay	Ontario	OK558805
ON10	Symptomatic	Pinot noir	Ontario	OK558806
ON11	Asymptomatic	Pinot gris	Ontario	OK558807
ON12	Asymptomatic	Mourvèdre	Ontario	OK558808
ON13	Asymptomatic	Pinot gris	Ontario	OK558809
BC2	Asymptomatic	Riesling	BC	OK558810
BC3	Asymptomatic	Merlot	BC	OK558811
BC4	Asymptomatic	Pinot gris	BC	OK558812
BC5	Asymptomatic	Cabernet franc	BC	OK558813
BC6	Asymptomatic	Cabernet franc	BC	OK558814
BC7	Asymptomatic	Cabernet franc	BC	OK558815
BC8	Asymptomatic	Cabernet franc	BC	OK558816
NS1	Asymptomatic	Vidal blanc	NS	OK558817
QC-1	Asymptomatic	Vidal blanc	Quebec	OK117409
QC-2	Asymptomatic	Vidal blanc	Quebec	OK117410
QC-3	Asymptomatic	Vidal blanc	Quebec	OK117411
QC-4	Asymptomatic	Vidal blanc	Quebec	OK117412
QC-5	Asymptomatic	Vidal blanc	Quebec	OK117413

**Table 3 viruses-15-00735-t003:** Sequencing statistic of grapevine Pinot gris virus isolates in this study.

Sample	Average Read Lengths	Total Reads	Reads after Trimming	Reads after Grapevine Genome Subtraction	Number of Reads Mapped to GPGV	%GPGV Coverage	Average Depth
ON1	121.25	1,404,086	1,384,766	33,084	1120	99	18.74
ON2	136.78	1,641,192	1,514,868	72,833	825	100	15.14
ON3	133.95	1,726,788	1,700,030	55,504	584	100	10.44
ON4	135.17	2,137,118	2,077,197	126,740	1365	100	24.8
ON5	132.41	1,353,302	1,320,808	95,302	2182	100	39.72
ON6	134.57	1,330,924	1,279,528	125,926	1451	100	26.76
ON7	129.02	1,631,786	1,608,789	81,286	1081	99	19.13
ON8	129.68	1,959,060	1,941,252	503,936	5386	100	95.52
ON9	133.35	1,497,350	1,415,854	70,292	719	99	12.9
ON10	134.02	1,908,282	1,852,319	44,348	2012	100	36.67
ON11	135.25	1,751,528	1,751,528	124,716	2493	100	46.16
ON12	132.77	1,102,282	1,038,856	66,460	1603	100	28.68
ON13	137.77	1,945,538	1,893,454	72,272	969	99	17.92
BC2	136.86	1,613270	1,562,264	16,897	726	99	11.51
BC3	119.06	701,520	679,734	10,466	513	98	9.36
BC4	118.28	629,980	614,434	23,472	812	97	13.25
BC5	122.31	718,630	666,548	18,262	313	96	5.19
BC6	121.05	598,336	553,412	11,496	212	95	4.45
BC7	122.03	486,514	447,910	9520	354	96	5.37
BC8	118.14	569,908	555,331	34,894	732	99	12.23
NS1	116.92	597,264	580,382	26,796	1055	99	16.89
Average	119.085	1,000,675	982,574	29,940	1087.5	99	17.815

**Table 4 viruses-15-00735-t004:** Other viruses and viroid species detected in the 21 libraries surveyed using Virtool–PathoScope and their genome coverage, respectively. Acronyms: hop stunt viroid: HSVd; grapevine rupestris stem pitting associated virus: GRSPaV, grapevine associated tymo-like virus: GaTLV; grapevine leafroll-associated virus 3: GLRaV-3, grapevine yellow speckle viroid: GYSVd, grapevine Syrah virus 1: GSyV-1, grapevine fleck virus: GFkV, grapevine red blotch virus: GRBV, grapevine leafroll-associated virus 1: GRLaV-1, grapevine virus T: GVT.

Library	HSVd	GRSPaV	GaTLV	GLVR-3	GYSVd	GSyV-1	GFkV	GRBV	GLRV-1	GVT
ON1	88	37.3	99	0	0	0	0	0	0	0
ON2	0	30.3	0	0	100	0	0	0	0	0
ON3	0	0	99	0	98	0	0	0	0	0
ON4	0	0	0	0	100	0	0	0	0	0
ON5	100	0	100	0	62	0	0	0	0	0
ON6	100	0	100	0	0	0	0	0	0	0
ON7	0	22	100	0	0	0	0	0	0	0
ON8	100	22	100	0	0	0	0	0	0	0
ON9	100	65	99	0	0	0	0	0	0	0
ON10	100	66	0	0	0	0	0	70	0	80
ON11	100	38	44	0	0	15	0	0	0	0
ON12	100	32	45	0	0	0	0	0	0	0
ON13	100	80	0	94	0	0	0	0	0	0
BC2	100	52	81	64	0	0	30	0	91	0
BC3	100	67	0	0	0	20	55	0	0	0
BC4	100	65	0	100	0	0	0	0	0	0
BC5	100	75	0	100	0	0	0	0	0	0
BC6	98	85	0	47	96	0	0	0	0	0
BC7	90	47	0	46	0	0	0	0	0	0
BC8	100	63	0	99	0	0	0	0	0	0
NS1	100	60	0	100	0	0	0	0	0	0

## Data Availability

All of the sequence data generated in this study are deposited in the NCBI Genbank, and the verified accession numbers are provided in the tables in the manuscript as well as in the Appendix A.

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
