# Peer review of "Phylogenetic and Evolutionary Studies of Grapevine Pinot Gris Virus Isolates from Canada"

_viruses, 2023, doi:10.3390/v15030735_

Round 1

Reviewer 1 Report (Previous Reviewer 2)

Second Round of revision - Manuscript viruses-2182246

The revised version of the manuscript entitled Phylogenetic and evolutionary studies of Grapevine Pinot Gris Virus (GPGV) isolates from Canada” resulted significantly improved compared with the previous version. Authors have greatly expanded the meaning of their work contextualizing and discussing their results on the light of more recent discovery. I have really appreciated the detail description about the presence of additional viruses and viroids in grapevine plants objected of this study. This data represents an important information also for other virologist which are involved or that study GLMD disease. Evolutionary analyses, including detection of recombination events, are now well described, clear and concise. Results obtained in this study strong support the discussion. 

However, I still found some minor mistakes, typing errors, and poor english, that I reported below.

Thus, before accepting the publication of this work in Viruses, authors should provide a new revised version, which must be subjected to extensive english revision (IT IS MANDATORY).

FOR THE AUTHORS: The symbol “>” means “change the sentence in the manuscript as followed”

Minor comments:

Why the format of the page is in A4 landscape?

Authors should revise name of the viruses mentioned in the manuscript, according to the ICTV nomenclature (please, refer to https://ictv.global/about/code)

Line 14: provinces > regions

Line 16: Our full genome-based phylogenetic analysis > In this study, phylogenetic analysis based on full genome sequences revealed…

Line 17: of European and Asian origin > From Europe and Asia 

Lines 17-19: Sentence is unclear, English must be revised.

Lines 19-20: involving 169 isolates representing 14 countries and four continents > involving 169 isolates from 14 different countries worldwide.

Line 20: distinctive clades with little regard to their country > distinct clades regardless the country.

Lines 21-22: with clade 1 containing majority of asymptomatic isolates (81% asymptomatic) and clade 2 containing predominantly symptomatic isolates (78% symptomatic) > Clade 1 included the majority of asymptomatic isolates (81% asymptomatic), while clade 2 was predominantly formed of symptomatic isolates (78% symptomatic)

Line 22: origin > evolution

Line 37: [2]The.. > add space.

Line 38: Remove (RAPs)

Line 38: (CP) respectively > (CP) genes, respectively

Line 40: although not on berries > although GPGV was never found to cause any symptoms on berries.

Line 41: [31],the > add a space

Line 42: prompting a number of studies on the origin and symptomology of its strains [32][33] > prompting a number of studies on the origin and evolution of the virus [32] SPACE [33].

Line 43: GPGV is known to be graft transmissible among Vitis species > where the authors found this information? Please, add a proper citation.

Line 44: In the Trentino region of Italy, for example, cultivar Teroldego appears to be resistant to GLMD > where the authors found this information? Please, add a proper citation.

Lines 45-46: Authors still insist about the severe symptoms displayed by Pinot gris and Traminer, while in the literature, as well as in Saldarelli et al., 2015 already cited by the authors, the cultivars that may show the most severe symptoms are Tocai/Friulano and Glera!!! (Please, also refer to Bianchi et al, 2015). Authors should mention these cultivars in the text. 

Line 50: Due to the uncertainty in its symptomology > Due to the uncertain symptomology

Line 52: One of the earliest studies reported a mutation in the stop codon of the MP gene which shortens the movement protein by 6 amino acids > it is not a mutation; it is a non-synonymous single nucleotide polymorphism (ns-SNP) that causes a nucleotide change (Thymine > Cytosine), replacing glutamine > stop codon. Please modify the sentence as followed: One of the earliest studies reported the presence of a non-synonymous single nucleotide polymorphism (ns-SNP) at the 3'-end of MP gene of the virus, which causes a nucleotide change, from Cytosine into Thymine, substituting a glutamine residue with a stop codon. Glutamine was usually present in MP gene of GPGV isolates from asymptomatic grapevines (latent), while stop codon was commonly detected in MP gene of GPGV isolates from asymptomatic grapevines (virulent), which showed a shortens of the MP gene by 6 amino acids.

Line 53 and onward: change mutation with nucleotide substitution or polymorphism.

Lines 57-61: Furthermore, in 2019, a full genome analysis of 20 GPGV isolates extracted from a variety of cultivars (Pinot gris, Glera, Tocai Fiulano, Tannat, Merlot, Riesling, Silene latifolia, Vetliner and Touriga nactional) with various countries of origin (Italy, Uruguay, France, Canada, USA, Germany and Slovakia) revealed that they were segregated into at least two groups, one of which is exclusively comprised of asymptomatic isolates [33] > That is incorrect. Tarquini and colleagues sequenced the full genome of 20 GPGV isolates from different ITALIAN cultivars; GPGV genomes collected from cvs Tannat, Merlot, Riesling, Vetliner and Touriga nactional were retrieved from NCBI; These genome sequences were produced by other researchers, not by Tarquini and colleagues); Moreover, Silene latifolia is an herbaceous host. Please revised the sentence properly. 

Line 61: that the involvement > that the PUTATIVE involvement

Line 63 and onward: Authors should specify which end of the MP gene. Please, replace “in the end of the MP region” with “at the 3’-end of the MP gene”.

Line 67: the end portion of MP gene plays a crucial role. > the 3’-end of MP gene MAY play a crucial role.

Line 208: may have been recombination events between them > recombination events between isolates included in the same clade may occur. 

Line 211: FEM01 is an herbaceous host; please, includes this info in the text.

Lines 214-215: Authors should show exclusively the ML tree deduced by the largest recombination-free fragment (4866bp), reported in figure 2S-B, since figures S2 (A and C) are poorly informative.  Thus, figures S2 A and C should be removed, maintaining exclusively the figure 2S - B. Authors should enlarge the remaining ML tree, revising properly the caption.

Line 215: Figure 2A, B and C > Figure S2 (maintain exclusively figure S2-B); Move supplementary Figures S2 after supplementary Table S5

Line 228: Modify caption and text of figure 2 properly (according to above-reported revisions)

Line 234 and onward: replace “representing” with “including” or “consisting of” or “clustering”. 

Line 236: are > were

Line 237: is > was

Lines 244-247: This part about the ratio of GPGV isolates found in symptomatic Versus asymptomatic grapevine is really interesting; However the sentence was not well written, resulting unclear. Authors should explain better the results. 

Lines 253-254: Red highlights indicate symptomatic isolates > GPGV isolates collected from symptomatic grapevine have been written in red.

Lines 267-268: The alleged virulent isolate fvg-Is12 was found in Clade β, while the latent fvs-Is15 was found in clade α (Figure 3) > “alleged” is not correct if authors referred the term to the symptoms. Symptoms were clearly detected on infected grapevines. Please, modify the sentence as followed: The fvg-Is12 isolate collected from symptomatic Pinot gris grapevine was found in Clade β, while the fvs-Is15, which was isolated from asymptomatic Pinot gris grapevine, was included in clade α.

Line 276: GPGV has been discovered 11 years ago, not so recently... Please modify.

Line 280: genetic variation is the dominant factor in symptom expression > genetic variation may be involved in symptom expression...

Line 281: 2018[31] > add space

Line 283: (4.6%.)of > add space

Line 292: [37]and > add space

Line 296: The relationship between GPGV isolates in Canada and GPGV isolates around the world > The relationship between GPGV isolates identified in Canada and those reported worldwide.

Line 297: by phylogenetic analysis of 64 GPGV isolates from eight countries and three continents > by phylogenetic analysis. 64 GPGV isolates from eight different countries and three continents were included in the study.

Line 298: Since viruses frequently engage in evolutionary processes that might violate the assumption of simple sequence substitution model > ADD A PROPER CITATION

Line 302: provided similar > backspace

Line 312: repeatedly[45][46] > add space

Line 315: species[24][49] > add space

Line 372: Demonstrated by Tarquini et al in 2021, infected vine can show symptoms at first and become completely symptomless after 4 months post inoculation. That’s right but the paper was published in 2019 (Tarquini, G., et al. 2019. Agroinoculation of Grapevine Pinot Gris Virus in tobacco and grapevine provides insights on viral pathogenesis. PLoS One, 14(3), e0214010). Authors should explain that in their work Tarquini and colleagues employed infectious clones of the virus (Fvg-15 or latent, and Fvg-12 or virulent), inoculating virus-free grapevines by the root (agrodrench) in controlled conditions (greenhouse). This is an important aspect because Tarquini and colleagues artificially reproduced GLMD disease, which partially mimicked the natural infection occurs in vineyard. 

Line 390: conclusion section results scarcely informative. Authors should improve the content or remove it.

Minor comments in figure captions:

Figure S1: Depth and coverage of genome sequencing of Grapevine Pinot gris virus from 21 libraries comprised with CGW

Table S3 and S4: that were detected using by seven methods using > “using” was reported twice; please find a synonymous. 

Table S5: Bold font represent segments with 200bp or longer > Recombination-free genome fragments 200-bp sized or longer were reported in bold.

Minor comments in Reference: the literature style is not consistent with that of the journal. Please, revise.

Author Response

The authors revised all virus names according to ICTV nomenclature. The referencing style has been adjusted in accordance with the Journal. Suitable references and further clarifications were added in places that the reviewers suggested. All grammatical errors and typos were also corrected.

Reviewer 2 Report (New Reviewer)

Manuscript viruses-2256927 describes the incidence, genetic variability, and phylogenetic relationships of Canadian isolates of grapevine Pinot gris virus. The research is of interest and thoroughly conducted, and the conclusions are justified by the findings. In addition, the manuscript is overall well written, although editorial changes should be considered to improve the quality of the text.  See suggestions below:

Line 2: Change Grapevine Pinot Gris Virus to grapevine Pinot gris virus

Line 2: Eliminate (GPGV)

Line 13: Change Grapevine to grapevine

Lines 15-16: … 43 GPGV isolates from eight countries and …

Line 19: Change representing to from

Line 20: … containing the majority …

Line 22: Eliminate attempt

Line 27: Change Grapevine Leaf Mottling and Deformation to grapevine leaf mottling and deformation

Line 29 and throughout the manuscript: Change Grapevine Pinot gris Virus to grapevine Pinot gris virus

Line 30: Add a space between found and [2].

Line 30: … GPGV’s causal role…

Line 31: Vitis vinifera should be italicized

Line 31: Add a space between Algeria and [4], and between Armenia and [5]

Line 32: Add a space between Australia and [7]

Line 33: Add a space between Greece and [16], and between Middle East and [19]

Line 34: Add a space between Czech Republic and [23], Spain and [26], and Turkey and [27].

Line 36: Italicize Trichovirus and Betaflexiviridae

Line 37: … [2]. The viral genome …

Line 37: Add a space between 7.250 and bp

Line 37: Change 3 to three

Line 38: … (CP), respectively

Line 39: Change Grapevine to grapevine

Line 42 Change [32][33] to [32,33]

Line 43: GPGV is graft transmissible …

Line 43: Italicize Vitis

Line 43: Change symptoms to symptom

Line 46: Change [29], [34] to [29,34]

Line 49: Change [29], [35] to [29,35]

Line 50: Eliminate its

Line 58: Eliminate extracted

Line 59: Change national to Nacional

Line 60: Eliminate were

Line 63: Add a space between 356 and bp

Line 64: Change a non-symptom-inducing to asymptomatic

Lines 69 and 81: Change for to of

Lines 69-70: … isolates, respectively.

Line 76: Add a space between Columbia (BC) and [11], Quebec (QC) and [38], and Nova Scotia (NS) and [39].

Line 78: … relative incidence of GPGV in ON and BC, and to investigate …

Line 85: …in the respective …

Line 89: … regardless of the …

Lines 92-93: … was calculated by dividing the number of positive samples in RT-PCR by the total number of samples.

Line 94: Change twenty-one to 21

Lines 94-95: Italicize V. vinifera

Line 96: Both sides of what?

Line 105: Change 21 to twenty-one

Line 112: Eliminate resulted

Line 112: Change viroid to viroids

Line 124: Grapevine genome sequences were eliminated …

Line 128: Change 5 to five

Lines 128-129: this sentence reads oddly. Should it read as: Additionally, five assembled genomes of GPGV were provided by Dr. Fall’s lab at …

Line 134: The 64 GPGV full-length…

Line 140: … full-length GPGV genome …

Line 143: … the 64 full-length genome sequences …

Line 144: Change Apple to apple

Line 145: Italicize Trichovirus

Line 147: Add a space between 4866 and bp

Line 153 Add a space between 200 and bp

Line 155: Change in to at

Line 157: Change for to of

Line 157: Add a space between 460 and bp

Line 169 and throughout the manuscript: Change Sauvignon Blanc to Sauvignon blanc

Line 169 and throughout the manuscript: Change Pinot Noir to Pinot noir

Line 170: Change 3 to three

Line 171: Change Gamay Noir to Gamay noir

Line 171 and throughout the manuscript: Change Cabernet Franc to Cabernet franc

Line 171 and throughout the manuscript: Change Baco Noit to Baco noir

Line 185: Change 5 to five

Table 3, caption: Change Grapevine to grapevine

Line 189: Change 8 to eight

Lines 189-190: Change Grapevine to grapevine

Line 191: Change Hop to hop

Line 191: Change Grapevine to grapevine (twice)

Line 192: Change Grapevine fleck virus to grapevine fleck virus

Lines 199-201: The first name of viruses and viroid should not be capitalized

Line 213: Add a space between 200 and bps and change bps to bp

Line 214: Add a space between 200 and bp

Line 216: Add a space (between 866 and bp

Lines 222-225: What do the branches circled in red represent?

Line 234: Change 2 to two

Line 260: Change 2 to two

Line 277: Change [11], [39], [38] to [11,38,39]

Line 277: Chane Despite that to However

Line 278: Eliminate that focused

Lines 281, 282, 285, and 286: Change indicent to incidence

Line 282: Add a space between 2018 and [31]

Line 289: Change viroid to viroids

Line 292: Add a space between [37] and and

Line 293: … leafroll -ssociated …

Line 310: Change Betaflexiviridea to Betaflexiviridae

Line 311: … repeatedly [45,46].

Line 312: Add a space between vector and 47].

Line 314: Eliminate to

Line 315: … species [24,49]

Line 319: … full-length genome …

Line 320: Change bootstraps to bootstrap

Line 321: Change topology to topologies

Line 321: Eliminate of the

Lines 321 and 322: Change contain to containing

Line 325: … Tarquini et al. in …

Line 328: Eliminate J. M.

Line 328: Change was to were

Line 329: Eliminate Comparing to their study,

Line 332: … full-length genome …

Line 333: Change clade 2, 3 and 4 to clades 2, 3, and 4 …

Line 335: Hily et al.

Line 339: Eliminate indeed

Line 350: … clades [33, 35].

Line 354: Change 5 to five, and change 2 to two

Line 356: …. 80% of them …

Line 359: Change objected to tested

Line 361: Change for to of

Line 363: Change variance in to variability

Line 365: Change “wrong” to unexpected

Line 368: Eliminate has

Line 369: Change [30], [43] to [30,43]

Line 369: Eliminate in the samples

Line 372: Change vine to vines

Lines 372-373: Add a reference

Line 374: Change virus and viroid to viruses and viroids

Line 375: Eliminate other viruses and viroid uch as

Line 375: Change Grapevine to grapevine

Line 376: Change [39], [44], [50] to [39,44,50]

Line 376: … on a few previous studies [28,3551], …

Lines 378-379: Given that HSVd and GRSPaV were previously found to have no association with symptom expression [35], [51] … This statement in incorrect as GRSPaV can induce symptoms in Vitis rupestris. Please re-write this sentence for correctness.

Line 379: Change [35], [51] to [35,51]

Line 381: Change consistence to consistent

Line 385: Eliminate with

Line 386: Change GLFV to GFLV

Line 388: Change symptoms to symptom

Lines 393-394: Eliminate the analysis focused on the MP/CP region showed that

Lines 397, 400, 402, 404, 407, 414, 416, 419, 421, 422, and 424: Change Grapevine to grapevine

Line 413 Add a space between 200 and bp

Lines 443-451: Are these two paragraphs needed?

Supplementary table S1: Change Grapevine Pinot Gris Virus to grapevine Pinot gris virus

Supplementary table S1: Change Sauvignon Blanc to Sauvignon blanc

Supplementary table S1: Change Pinot Gris to Pinot gris

Supplementary table S2: Change Cab Franc to Cab franc

Supplementary table S2: Change Pinot Noir to Pinot noir

Supplementary table S2: Change Vidal Blanc to Vidal blanc

Supplementary tables S3, S4, S6, S7, S8, S9, S10, and S11: Change Grapevine to grapevine

Author Response

The authors revised all virus names according to ICTV nomenclature. The referencing style has been adjusted in accordance with the Journal. Suitable references and further clarifications were added in places that the reviewers suggested. All grammatical errors and typos were also corrected.

This manuscript is a resubmission of an earlier submission. The following is a list of the peer review reports and author responses from that submission.

Round 1

Reviewer 1 Report

Review sheet regarding the article from Minh Vu et al. submitted to viruses entitled: Symptomatology and phylogenetic relationships of Grapevine Pinot Gris Virus isolates from Canada.

The manuscript tries to provide information and shed some light on the origin, genetic diversity of Canadian GPGV isolates as well as the potential link between genetic variants and symptomatology. Unfortunately, based on the manuscript provided, I can question the significance, the relevance and originality and the reliability of the study. This manuscript is unfortunately not well written, full of errors, spelling errors and many inaccuracies. Some parts are more speculative than scientifically solid. Even the title is misleading. How can the authors decipher the relationship between symptomatology and sequences diversity from Canadian isolates, when out of the 21 isolates being used, only one is symptomatic?

Many problems are scattered along the paper, with some of them, but probably not all of them, being listed below.

1.       Line 32: when the author mention ‘the origin’, are they talking about the Origin of the virus worldwide or origin of the virus in Canada.

2.       Line 41: to my knowledge, we can not define GPGV as the causal agent of GLMD yet. Koch postulates have not been completely fulfilled.

3.       Line 55: presence of GINV is not only confirmed in Japan. Its presence has been confirmed in China, directly form samples (Fan et al., 2017 in Pl. Disease) and indirectly via datamining (Hily et al., 2020 in PhytoB. J.).

4.       Genetic Variance? Do the authors mean Genetic diversity?

5.       While the goals of the study are clearly stipulated and enunciated (line 93-96), the means to reach them are far from being sufficient to reach them

6.       Line 107: details about the TNAs extraction protocol are needed (paper…).

7.       Line 109:while primers from the study [34] have been used in particular? While not use other primers designed to englobe the  whole known genetic diversity so far described in Hily et al. 2021, PhytoB. J.?

8.       Line 118: total RNAs were….

9.       Line 128: Brock Univ., Canada

10.   Line 128: more info needs to be provided about reads… are they paired end? Size? ….

11.   Line 140: what if those 1000 reads map on 100 nt of a 10 000nt genome, is this can be considered a positive sample?

12.   Line 153-153: can the authors provide the parameters used for mapping? Read length and similarity?

13.   Line 155 and 156: I do believe it’s, for both references, Table 2 instead of Table 1.

14.   2.3.3 Recombination analyses: can the authors better explain table S3? What is ‘even number’ means? Why referring only 3 lines instead of 6 mentioned in the text? Also, can the authors mention the name of the sequences?

15.   Line 166-167 is not properly written.

16.   Line 184 and all along the manuscript. Can the authors define what is a ‘clade’? what is the ‘cut-off’ ? If so, can the authors explain that in table S4 and S5 an intra-clade mean distance (clade 4) can be bigger than the lowest inter-clade (clades 1 vs 2) mean distance?

17.   Lines 188-191. Sentence not very scientific. What are those ‘important’ information. If the authors refer the the SNP at the end of the MP, then this has been refuted by many labs, as well as the original team (Marra et al., 2020. J. of Pl. Pathology).

18.   Line 220-222. Looking at Figure S1, I am not convinced that complete genomes were recovered from all 21 samples. BC3, and others, did not seem to have any coverage around the 5200-5300 nt. Is an average coverage of 4 or 5 is enough to recover the full GPGV sequence? Also, can the authors explain the different gradient of blue color?  

19.   Section from lines 228 to 234 is poorly written.

20.   Line 232: to my knowledge, GaTLV is an environmental virus of grapevine but not a grapevine-infecting virus.

21.   Line 233 to 234: is this part of the paper? Is it a draft or the final version of the paper?

22.   Table 4: to me, coverage %age and or read # are not sufficient in order to define the presence of a virus. RPKM seems to be more relevant to describe infection status.

23.   In Table 4 and all along the paper: many acronyms of viruses are wrong. The same is true for virus names all along the paper. The authors need to be more rigorous.

24.   Line 239: from how many sequences and which one, those 19 recombination events have been predicted? By the way, have they been indeed ‘detected’ and confirmed by molecular biology means or is it only using RDP4? If so, it is only a prediction.

25.   Table 5: can the authors give the cut-off for a recombination events to be (+) ?

26.   Line 246 to 249 is poorly written.

27.   Lines 249-250: if your selection from the beginning is only American isolates, indeed, in the end, your clade represented will only be from North America. To properly do the experiment, the authors need to upload and use all full GPGV sequences available in the public databases. If the authors use only partial data, they cannot properly conclude.

28.   Figure 1, 2 and 3. I am not sure that the cladistic view is the best way to show the results, especially after using a powerful model such as ML. To properly conclude, distance should be included, clade definition should be properly stipulated. Also, not only nodes above 70% are shown, but many of them below are also represented. Out of this work, no proper conclusions can be draw.

29.   Lines 291 to 293. How the authors can concludes while having only 11 sequences from symptomatic plants vs 47 from asymptomatic. The same remark is true regarding the mention of clade 3 and 6 with only 2 and 3  sequences respectively.

30.   Line 324: does the authors have a study showing the fact that GPGV is of economic importance?

31.   Lines 331 to 338 is not a discussion but a repeat of the result section.

32.   Lines 349 to 351 is poorly written.

33.   Lines 373 to 378. On can the authors state such things? Based on which hypothesis? If from using the full length of the genome, maybe, if from a SNP, the methodology is not powerful enough to discriminate it.

34.   Lines 391 to 394. Very poorly said.

35.   Lines 395 to 397: I do believe that the virus titer has been refute as well as the SNP in the MP.

36.   Line 403: why authors are talking about GVA and GVB? The data never mentioned any of these viruses

37.    Line 405: what the reference [44] is referred here? Is this comment about GPGV or GRBV?

38.   Line 410 to 411. So if I read well the line, symptoms are not due to GPGV but GRBV and GVT?

39.   Line 419: from where the # 50 is coming from?

40.   Conclusion. I don’t know where from this study, the authors have provided new insight about the origin of GPGV? not even in North America nor Canada? The last sentence (lines 426-428) is simply not true.

Reviewer 2 Report

Manuscript ID: viruses-2182246 entitled “Symptomology and phylogenetic relationships of Grapevine Pinot gris virus isolates from Canada” describes genetic variability of GPGV isolates collected from either symptomatic or symptomless grapevines grown in different Canadian regions. In silico sequence analyses were conducted in order to evaluate the origin/evolution of Canadian GPGV isolates and to attempts to establish phylogenetic relationship(s) with those reported in other countries. The authors also tried to establish a sort of correlation between phylogenetic data and symptoms expression and/or severity, unfortunately without success.

Despite the study has been well conducted, it sounds poor attractive and scarcely innovative, mostly because the authors have mentioned much old and not update literature, without referring to the recently published data.

Tarquini and co-authors produced many studies that clarified the role of GPGV as aetiological agent of GLMD, also EXPERIMENTALLY demonstrating the involvement of genetic variability (i.e. non-synonymous SNPs, ns-SNPS) detected at the 3’-end of MP gene in symptom severity, virus titre, and antiviral defence (Polymorphisms at the 3’end of the movement protein (MP) gene of grapevine Pinot gris virus (GPGV) affect virus titre and small interfering RNA accumulation in GLMD disease; Tarquini et al., 2021). Moreover, in 2022 a detailed review was also published by the same group: The conundrum of the connection of grapevine Pinot gris virus with the grapevine leaf mottling and deformation syndrome, Tarquini et al., 2022).

On this regard, in order to consider the manuscript for its publication in Viruses, some major points (and some minor revisions) have to be amended:

1) The authors have to update the literature cited and revise their results in light of the recent knowledge provided by Tarquini and co-authors (2021 and 2022), before mentioned.

2) The presence of recombination breakpoints (RBs) may affect phylogeny reconstruction. However, recombination plays a pivotal role in virus evolution/adaptation, and it could be associated, or it may suggest the onset/incoming of novel/emerging viral strain (probably with augmented virulence, (“Analysis of new grapevine Pinot gris virus (GPGV) isolates from Northeast Italy provides clues to track the evolution of a newly emerging clade”, Tarquini et al., 2019). In case of some RBs are detected along viral genomes, a proper “mode of action” is to “remove” the genomic locus(es) where RBs are reported (Tarquini et al., 2019; Hily et al., 2020). Instead, the authors excluded viral isolates from the analysis, losing a lot of information about the origin and/or the evolutionary history of GPGV in Canada.  Authors have to repeat phylogenetic studies twice, including ALL the GPGV isolates in the analysis. A first investigation should include the full genome sequences of GPGV isolates objected of this study. Then, a second analysis should include only partial genome sequences of all GPGV isolates, with exclusion of regions where RBs were detected (see Tarquini et al., 2019, pp.3 “A preliminary cluster analysis consisting of the construction of a neighbour network… [16]), showed a significant reticulation, indicating the presence of contrasting phylogenetic signals (Fig. 1a). Because recombination can be a serious confounding factor for phylogeny reconstruction [17], we searched for evidence of recombination in the GPGV sequence using GARD (Genetic Algorithm Recombination Detection [18]) and RDP4 (Recombinant Detection Program [19]), and highly significant recombination breakpoints were detected by all tested algorithms at nt positions 788 and 2208; a further breakpoint at nt position 2811 was detected by GARD, although its significance was not supported by all algorithms tested (Table S2). Based on these findings, the genome sequence alignment was split into four segments. The largest segment, which spans 4400 nucleotides from the putative recombination breakpoint at nt position 2811 to the 3’end of the viral genome, allowed informative phylogenetic reconstructions to be made (Fig. 1B, and Fig. S2). 

3) Many authors reported the presence of ns-SNPs at the 3’-end of the MP gene of the virus, suggesting their putative role in expression of GLMD symptom and their severity (Saldarelli et al., 2015; Bertazzon et al., 2017; Tarquini et al., 2019a). In 2021, Tarquini and colleagues developed a chimera-GPGV clone that was agroinoculated in virus-free Koober rootstock, experimentally demonstrating, for the first time, the pivotal role of sequence variability (i.e. ns-SNPs) in GLMD severity (symptom expression, virus titre, Boron uptake, vsiRNA production). The chimeric clone was obtained by replacing the 3’-terminus of the MP gene of a GPGV infectious clone derived from an isolate from grapevine showing severe symptoms (i.e. fvg-12), with a 356 bp synthetic DNA fragment having a sequence resembling that of another GPGV isolate (i.e. fvg-15), recovered from an asymptomatic grapevine. In order to better clarify the role of genetic variability in symptom expression, the authors should assess the presence of these (o different) ns-SNPs at the 3’-end of MP gene of GPGV isolates objected of this study, in the context of symptom expression/severity.

4) The authors have diagnosed additional viruses and viroids in GPGV-infected grapevine. Previous studies demonstrated that the presence of some of them was not significant in GLMD disease, since they didn’t affect symptom expression or disease severity (Bianchi et al., 2015; Bertazzon et al., 2017; Tarquini et al., 2018, 2019a; 2019b; 2021). Otherwise, the putative role of some “less-common” viruses, such as those that was never detected outside American countries (i.e. European countries), in GPGV infection is poorly understood. In this regard, authors have to provide more convincing evidence (literate studies are enough) or delete all association between co-infection and GLMD symptoms along the entire manuscript. 

In general, authors should avoid associating symptom expression/severity with genetic variability (i.e. lines 259-262; 271-274; 291-294; 313-315) since no evidence are available so far. Also, the authors should re-think the title, avoiding the use of the terms “symptoms/symptomatology” that is too much speculative. NO EVIDENCE was found in the manuscript that allow to clearly correlate phylogenesis (or geographical origin) with GLMD symptom. 

Minor revision: 

-        The title has to be changed; I suggest: “Phylogenetic and evolutionary studies of Grapevine Pinot Gris Virus (GPGV) isolates from Canada” but the authors are free to modify the title as their prefer. Please, avoid any type of link with symptoms.

-        Lines 16-17: Change this sentence “This study investigated the correlation between genotype and symptomology of Grapevine Pinot gris virus (GPGV) isolates from Canada with GPGV strains that were reported from around the world”, with ““This study investigated the phylogenetic relationship between Grapevine Pinot gris virus (GPGV) isolates from Canada with GPGV strains that were reported from around the world”.

-        Line 18: Change “all four grape-growing”, with “the main four grape-growing”

-        Line 24, 51,181 (and along the entire manuscript): replace “RNA dependent RNA polymerase (RDRP)” with replicase-associated proteins. RDRP is just one of the proteins involved in GPGV replication, together with Helicase and Methyltransferase (Giampetruzzi et al., 2012).

-        Lines 24-25: Remove this sentence “Individual gene analysis of the RNA dependent RNA polymerase (RDRP) gene, the movement protein (MP) gene and the coat protein (CP) gene, resulted in similar phylogenetic relationships”. 

-        Line 26: Delete this sentence “None of those analyses revealed a strong link between genotype and symptomology”.

-        Lines 41-42: In 2012 GPGV was just discovered by HTS; No evidence was reported, at that time, about the involvement of the virus in GLMD disease. Their role as causative agent of GLMD disease was revealed, for the first time, in 2019 (Agroinoculation of Grapevine Pinot Gris Virus in tobacco and grapevine provides insights on viral pathogenesis, Tarquini et al., 2019b) and further confirmed in 2021 (Trigger and Suppression of Antiviral Defenses by Grapevine Pinot Gris Virus (GPGV): Novel Insights into Virus-Host Interaction, Tarquini et al., 2021a; Polymorphisms at the 3’end of the movement protein (MP) gene of grapevine Pinot gris virus (GPGV) affect virus titre and small interfering RNA accumulation in GLMD disease, Tarquini et al., 2021b).

-        Line 50: Change “The viral genome is approximately 7.400 bp”, with “The viral genome is approximately 7.250 bp”.

-        Line 51: Replace “whose products are..”, with “whose encodes for..”

-        Line 53 (and along the entire manuscript): Please, refer to ICTV nomenclature when writing virus names. The name of the viruses mentioned in the manuscript are often wrongs (see below). Authors should use capital letter and italic when necessary.  

-        Lines 59-60: the sentence “In the Trentino region of Italy, for example, cultivar Teroldego appears to be resistant to GLMD” is inconsistent, authors should explain this statement in detail (or remove this part). In Italy, as well as in other countries, research have been focused on most commercially relevant cvs (i.e. Pinot gris, Pinot noir, Traminer, Merlot, Riesling..). For example, authors could cite the most sensitive Italian cv as Friulano (indicated as Tocai in Slovakia).

-        Lines 68-72: Regarding the statement “Until now, there have been a few attempts to associate symptomology with genetic variations of different GPGV strains through phylogenetic analysis. One of the earliest studies reported a mutation in the stop codon of the MP gene which elongates the movement protein by 6 amino acids [29]. This mutation was later found to be important but not decisive in symptom expression [29]”, should be re-write base on the recent study published by Tarquin et al., 2021. The presence of stop codon (and additional ns-SNPs) at the 3’-end of the MP gene was reported by many authors (Saldarelli 2015, Bertazzon 2017, Tarquini 2019), but the crucial role of such as SNPs (including the stop codon) in symptom espression and GLMD severity has been already experimentally demonstrated (Tarquini et al., 2021) and authors should take in consideration, along the entire manuscripts, this important achievement.

-         Lines 92-96: Please modify this sentence "Considering the economic importance and prevalence of GPGV, the objectives of this study were 1) understand the genetic variance in Canadian GPGV isolates, their origin, and their relationships with GPGV strains from other countries and 2) to gain further insights into the association between genomic characteristics and symptomology of GPGV", with "Considering the economic importance and prevalence of GPGV, the main objectives of this study were to investigate genetic variability of Canadian GPGV isolates, their origin, and their relationships with GPGV strains from other countries”.

-        Lines 99-100: This sentence “To estimate the incidence rate of GPGV in ON and BC, commercial vineyards were chosen randomly, and samples were collected proportionally from each vineyard to best represent the grapevine demographic in the region” is unclear. Please re-formulate. 

-       Line 105-106: Considering that samples were collected in September and October, which were reported as worst time for GPGV diagnosis (low virus titre in GPGV infected plants), authors should assess the viral titre in GPGV-infected grapevine objected of this study. They can use the reads abundance obtained via HTS. Moreover, the sentence should be replaced with “Samples were collected in September and October of 2020 and 2021, regardless the presence/absence, and severity of symptoms displayed by grapevine plants”.

-       Line 121: If it possible, authors should include the ratio of Abs 260/230 as also this value may affect the success of the HTS. 

-       Line 134-136: Please, include the quality parameter set for trimming.

-       Lines 166-168: Regarding the sentence “Sequences in recombination events that were detected with five or more methods were removed from downstream analysis”, and the major point num. 2 previously mentioned, authors should repeat the phylogenetic analyses properly as described above, maintaining all the GPGV isolates in the downstream analysis (or restricting the analyses to the sequence portions that are not affected by recombination). Comment(s) about the role of recombination in phylogenesis reconstruction should be included in the manuscript, as putative factor that interfere with symptom expression and/or GLMD severity. 

-       Lines 179-182: Please, replace the sentence "To determine if any viral gene was more crucial in symptom expression, we attempted to repeat the analysis for the 3 genes of GPGV, namely RDRP, MP and CP. However, the analysis for the CP gene could not be carried out since 22 out of 38 sequences obtained from NCBI did not contain a complete CP gene", with "Phylogenetic analyses were conducted on the full genome sequences, the replicase-associated proteins, and the MP gene of GPGV isolates objected of this study. Moreover, phylogenesis was also inferred from the most variable region overlapping the 3'-end of MP gene and the 5'-end of CP gene of the virus.

-       Line 186, 198 (and along the entire manuscript): The sentence “Codon positions included were 1st+2nd+3rd+Noncoding” is unclear. Please specify. 

-       Line 188: the first authors who identified the variable region and the stop codon in the MP/CP genes were Saldarelli and colleagues (2015). The authors that experimentally demonstrated the role of this region in symptom expression and GLMD severity were Tarquini and colleagues (2021). Please, replace the citations properly, for example: “Saldarelli and co-authors (2015) were the first that identified region of 460 nucleotides spanning across the MP and CP genes of the virus (base 6280th to base 6739th, NC_015782.1) which include several ns-SNPs and an additional stop codon in isolates collected from symptomatic grapevines. Recently, Tarquini and colleagues (2021) demonstrated that such as sequence variation really affected symptomology of GPGV”.

-       Line 209, Table 1: which is the meaning of GPGV +ve? Please explain in the caption.

-       Line 227, paragraph 3.3: Please, describe better the viruses and viroids detected in grapevine plants objected of this study (in addition to GPGV). The table reports more viral species than those listed in the text (i.e: GLRaV-1 – Grapevine leafroll virus 1; GYSVd – Grapevine yellow speckle viroid; GSyV-1 – Grapevine Syrah virus 1; GFV – Grapevine fleck virus, GVT – Grapevine virus T were not cited in the text). The name/acronym of some of viruses are wrong: Grapevine fleck virus is GFkV and not GFK; GLVR-3 and/or GLRV-1 are GLRaV-1 and GLRaV-3 (...) Please refer to ICTV nomenclature and revise the entire manuscript (tables include). The name/acronym of the viruses should be added in the caption, above the table, and the full name should be placed before the acronym (i.e. Grapevine leaf-roll associated virus-3, GLRaV-3 …).

About the involvement of these viruses in GLMD aetiology, please refer to major point num. 4 on the role of the spread viruses and viroids and their no-involvement in GLMD (Bianchi et al., 2015; Bertazzon et al., 2017; Tarquini et al., 2018, 2019, 2021).

-       Line 241: Modify “Table 4” with “Table 5”. 

-       Line 245, table 5: move acronym and name of bioinformatic tools in the caption, above the table.

-       Lines 254-262: Authors should report the name of the isolates after the clade number, so reader could distinguish GPGV strains, i.e. clades 1 and 2 contained six isolates from asymptomatic grapevines (names of the six isolates) and two isolates from symptomatic grapevine (names of two isolates). Moreover, viral isolates cannot be classified as “symptomatic” and “asymptomatic”. Symptomatic and asymptomatic were the host grapevines from which GPGV isolates came from. Please distinguish the GPGV isolates by indicating the presence/absence of symptoms on source plants (symptomatic/symptomless grapevine).

-       Figures 1, 2 and 3: to make the tree more understandable, authors should mark GPGV isolates from symptomatic plants (i.e. with a black dot) so readers can easily distinguish these isolates from those collected from symptomless plants.

-       Authors should include the description of results obtained by the analysis of replicase-associated protein in the same part of genome sequence analysis, obtaining one single paragraph, may entitle “Phylogenetic analyses of largest sequences: full genome and replicase-associated proteins”. 

-       Lines 259-262; 271-274; 291-294; 313-315: Authors should avoid any link to symptom expression/severity

-       Lines 296-302: caption of figure 2 should be include in caption of figure 1 (One paragraph describing both full genome and replicase-associated protein analyses).

The “new” figure 2 will be “Phylogenetic analyses of MP”, while the “new” figure 3 will be “Phylogenetic analyses of MP/CP”; Please, revise caption of respective new figures 2 and 3, properly.

-       Lines 328-330: The statement “determining whether genetic variation is the dominant factor in symptom expression has significant implications for disease management and propagative material sourcing” has been partially revealed (Tarquini et al., 2021); modify this sentence.

-       Lines 339-350: The role of multiple infection caused by different viruses needs to be explain in detail. Bianchi, Bertazzon, Tarquini are just some of authors that have already reported the non-involvement of certain viruses in GLMD symptoms. Otherwise, the putative role of viruses such as GRBV, GVT, GSyV exclusively detected in American vineyards, is still unknown. Authors should discuss this aspect, include more recent literature. 

-        Line 353: “We eliminated six isolates that shown strong evidence of recombination”; please revise this sentence properly, according with the major point num. 2.

-       Lines 379-392: this paragraph has to be modified according with previous comments. Please, include an updated literature in this section (including the role of MP/CP SNPs in symptom expression and GLMD severity, Tarquini et al., 2021).

-       Lines 396-399: please modify the literature properly. Several studies have investigated the trend of virus titre in GLMD severity.

-       Line 403: Grapevine fleck virus (GFkV), instead of Grapevine flake virus (GFkV)

-       Lines 405-417: Revise this section about mixed infection, according to the comments of major point num. 4.

-       Line 419: the viruses than can infect grapevine are 80 (almost 90) so far. Not 50.

Reviewer 3 Report

The manuscript from Vu and colleagues refer to Grapevine Pinot gris virus infections in Canadian vineyards. Beside reporting the incidence of the virus in three different provinces the study describes  the virome analysis of 21 selected sources and the phylogenetic comparison of their genome and/or genes with corresponding sequences available in the NCBI database.

The manuscript lacks a minimal description of the symptoms observed in the field. As this is a virus whose economical importance is a matter of discussion, the inclusion of these data, better supported by photos would be desirable. This argument assumes a further value for red cultivars for which the symptomatology is not well defined. Therefore I suggest the authors to include this data.

Few minor mistakes and clarifications are necessary: 

rows 155 and 156: Table 1 should be Table 2

row 240: Table 4 should be Table 5

chapter 3.5: please, include the reference to Figure 2 in the text

Caption to Figure S1: please change CGW with “CGW (CLC Genomics Workbench 20.0.4)”

Caption to table S3: Please, add in the caption “The event number is reported in Table 5”

Caption to table S4: Please, indicate what is below the diagonal. Distance?

Other corrections and suggestions:

row 38: Tretino should be Trentino

rows 53-55: change “The genome of GPGV shares significant similarities with Grapevine berry Inner Ne-crosis Virus (GINV) and both viruses evoke comparable symptoms on leaves and shoots 5 [26].” with “The genome of GPGV shares significant similarities with Grapevine berry Inner Necrosis Virus (GINV) and both viruses evoke comparable symptoms on leaves and shoots, although not on berries [26].”

rows 233-234: It is not clear to me the significance of the sentence “The text continues 233 here (Figure 2 and Table 2)”

row 324: I suggest not to be more direct on the economical importance of GPGV. This is jet a matter  of discussion. I suggest to weaken the sentence telling that “ it is a recently discovered virus whose economical importance is a matter of study”. Even this study does not give details about the symptomatology and the impact on the economy of Canadian viticulture.

As a general comment I noted that no considerations were done about the possible

involvement of nurseries or age of the vineyards in the spread of the viruses.

It seems from Table 1 that some cultivars (i.e. Riesling, Cabernet franc, Merlot) have  very diverse incidences, which, beside excluding the effect of cultivars in symptoms, points to a local spread, likely caused by the propagation material. In addition, if possible, some photos of symptoms, particularly for red varieties, and information about the economical importance (i.e. reduced production; less wood; smaller bunches) would be useful for the description of the GLMD. It is a pity that only a symptomatic isolate was included in the virome analysis although the study  report to the involvement of the MP/CP in the expression of symptoms although I agree that it is not the only determinant.

In conclusion the manuscript deserves its publication because of its updated of the Canadian GPGV incidence which has increased since the last studies, and further stimulates to perform additional recerchers to unravel the economical importance of this virus. As said before, the inclusions of symptoms details and photos would increase the impact and interest of the paper.

I therefore agree with its publication after minor revisions.